# Luxury Zinc Supply Prevents the Depression of Grain Nitrogen Concentrations in Rice (*Oryza sativa* L.) Typically Induced by Elevated CO_2_

**DOI:** 10.3390/plants12040839

**Published:** 2023-02-13

**Authors:** Niluka Nakandalage, Paul James Milham, Paul Holford, Saman Seneweera

**Affiliations:** 1Faculty of Veterinary and Agricultural Sciences, The University of Melbourne, Parkville, VIC 3010, Australia; 2Faculty of Agriculture, University of Ruhuna, Matara 81100, Sri Lanka; 3Hawkesbury Institute for the Environment, Western Sydney University, LB 1797, Penrith, NSW 2751, Australia; 4School of Science, Western Sydney University, LB 1797, Penrith, NSW 2751, Australia

**Keywords:** modified CO_2_ concentration, grain Zn, grain yield, tiller number, panicle number

## Abstract

Rice (*Oryza sativa* L.) has inherently low concentrations of nitrogen (N) and zinc (Zn), and those concentrations are falling as the atmospheric concentration of carbon dioxide ([CO_2_]) increases, threatening the quality of human diets. We investigated the effect of two levels of Zn supply (marginal and luxury), on Zn and N concentrations in whole grain of two indica rice cvv. Differing in Zn-efficiency (IR26 (inefficient) and IR36 (efficient)), grown in sand culture at ambient (400 µL CO_2_ L^−1^ (a[CO_2_])) and elevated (700 µL CO_2_ L^−1^ (e[CO_2_])) CO_2_ concentrations. For both cvv., luxury Zn-supply increased vegetative growth, and the foliar and grain Zn concentrations; the increases in grain yield were greater at e[CO_2_]. The e[CO_2_] decreased grain Zn concentrations ([Zn]), as is consistently observed in other studies. However, unique to our study, luxury Zn-supply maintained grain N concentrations at e[CO_2_]. Our data also show that enhanced Zn uptake is the basis of the greater Zn-efficiency of IR36. Lastly, luxury Zn-supply and e[CO_2_] appreciably decreased the time to panicle emergence and, consequently, to maturity in both cvv. Since Zn-supply can be manipulated by both soil and foliar applications, these findings are potentially important for the quality and quantity of the global rice supply. That is, further investigation of our findings is justified. Key message: Luxury zinc supply maintains grain N concentration at 700 µL CO_2_ L^−1^.

## 1. Introduction

Rice (*Oryza sativa* L.; Poales, Poaceae) is the most important staple crop throughout Asia, Latin America, and Africa [1]; it accounts for about 50–80% of the daily calorie intake of over half of the world population [2,3]. However, rice fails to meet the minimum daily intake requirements of zinc (Zn) [4,5]), iron (Fe) [6,7], and protein [8]. Rice productivity is generally expected to rise with the rising atmospheric concentration of carbon dioxide ([CO_2_]) [9,10], and through breeding for increased yield—both factors that decrease the concentrations of essential elements, such as Zn and protein (N) in the grain [11,12,13,14]. This depletion is exacerbated by consumer preference for polished rice [15], because polishing removes the outer, nutrient-rich grain layers [16,17,18]. That is, without intervention, rice is set to satisfy a decreasing proportion of human dietary requirements for Zn, Fe and N [19,20,21].

Inherently low concentrations of bioavailable Zn occur in many soils due to a variety of conditions [22], and are correlated with Zn-deficiency in rice and in humans [23,24]. In addition, rice cvv. vary in Zn-efficiency [25], and the bioavailability of soil Zn is readily manipulated agronomically [26,27]. Increasing the Zn-supply to wheat (*Triticum aestivum* L.) can increase the concentrations of Zn and N in the grain [28]. We tested for a similar effect in rice by growing two cvv. differing in Zn-efficiency in sand culture, at marginal and luxury Zn-supply, in combination with two atmospheric concentrations of CO_2_. The concentration of N in the medium was fixed.

## 2. Results and Discussion

### 2.1. Nutrient Status at Panicle Emergence

Panicle emergence is a time when the nutrient status of rice can be evaluated [29,30]. We used this opportunity to ensure that our plants were not affected by some unexpected mineral imbalance. The luxury Zn treatment increased the foliar concentrations of Zn ([Zn]) for both cvv. at both a[CO_2_] and e[CO_2_], and for corresponding treatments, the foliar [Zn] of IR26 was less than that of IR36 (Figure 1A). Within a cv. and Zn treatment, e[CO_2_] tended to increase foliar [Zn] and [N] (Figure 1A). The foliar concentrations of N ([N]) were 27,000–32,000 mg kg^−1^, and for a given cv. at both levels of Zn-supply, e[CO_2_] was associated with an increase in the [N]. This effect of e[CO_2_] on N was greater for IR26 at luxury [Zn] and at the marginal [Zn] for IR36 (Figure 1B). The concentrations of Ca, Cu, Fe, Mg and Mn were generally lower in IR26 than in IR36, and luxury Zn-supply decreased the concentrations of Mg, Mn, Fe and Cu (Appendix A). Nonetheless, in both cvv., all these essential elements remained within their respective sufficiency ranges 29,30]. That is, no known mineral imbalance compromised plant responses to the Zn and CO_2_ treatments.

### 2.2. Vegetative Growth

At panicle emergence, the biomass of IR36 was greater than that of IR26 receiving the same treatment (Figure 1A). For example, for IR26 at the marginal Zn-supply, the vegetative biomasses at a[CO_2_] and e[CO_2_] were 22.2 g and 24.5 g plant^−1^, and at the luxury Zn-supply were 42 and 53 g plant^−1^ (Figure 1A). For IR36, the corresponding biomasses at the marginal Zn-supply and a[CO_2_] and e[CO_2_] were 30 g and 38 g plant^−1^, and at the luxury Zn-supply were 50 and 60 g plant^−1^ (Figure 1A). With the luxury Zn-supply the biomass of both cvv. tended to increase, as did the response to e[CO_2_], but few of the effects were significant (*p* < 0.05, Figure 1A). Between panicle emergence and maturity, vegetative biomass continued to increase for both cvv., except for IR26 at marginal Zn-supply and a[CO_2_] (Figure 1B). At maturity, it is notable that luxury Zn-supply did not increase the vegetative biomass of IR36 at a[CO_2_] (Figure 1B). At no growth stage did plants exhibit foliar symptoms of Zn-deficiency in the Zn-sufficient treatment or of toxicity in the luxury-Zn treatment.

At marginal Zn-supply and at a[CO_2_] and e[CO_2_], the foliar [Zn] at panicle emergence for IR26 were 21 mg kg^−1^ and 25 mg kg^−1^, and for IR36 were 29 and 37 mg kg^−1^ (Figure 2). The greater [Zn] of IR36, combined with its greater biomass, translates into greater Zn uptake, which establishes the basis of the previously reported superior Zn-efficiency of IR36 [25]. This behaviour contrasts with the difference in Zn-efficiency due to different internal Zn requirements of two other rice cvv. [31]. That is, without additional information, high Zn-efficiency should not be assumed to necessarily be associated with a greater capability to acquire scarce Zn resources.

In our study, increases in both the Zn-supply and the [CO_2_] increased tillering, and at panicle emergence the relation was positive and linear for each cv. (*n* = 12):

For IR26, R^2^ = 0.59
Biomass (g plant^−1^) = 9.05 + 1.16 × tiller number plant^−1^(1)

For IR36, R^2^ = 0.57
Biomass (g plant^−1^) = 10.54 + 1.10 × tiller number plant^−1^(2)

Zn deficiency can decrease the number of tillers [32,33] and plant auxin synthesis [34]. Though these effects may be reasonably postulated at our marginal Zn-supply, Zn-deficiency is not a tenable explanation at our luxury Zn-supply [29,30].

Do the biomasses and the foliar [Zn] (Figure 1A and Figure 2; Equations (1) and (2) at panicle emergence collectively throw any light on the sufficiency of Zn-supply in the conventional plant nutrient sense? For Zn-inefficient IR26, the marginal Zn-supply was probably inadequate at both a[CO_2_], where the foliar [Zn] was 21 mg kg^−1^, and at e[CO_2_] where it was 25 mg kg^−1^) (Figure 1A and Figure 2). For Zn-efficient IR36, the biomasses indicate that the marginal Zn-supply may have satisfied its growth requirements at a[CO_2_] (foliar Zn 29 mg kg^−1^); although, based upon biomass at e[CO_2_], a foliar Zn concentration of 37 mg kg^−1^ was perhaps not sufficient (Figure 1A and Figure 2). By published standards established at a[CO_2_], all these foliar Zn concentrations at panicle emergence may have been sufficient to satisfy the growth requirements of rice [29,30]. The luxury Zn-supply produced foliar [Zn] for IR26 of 130 and 150 mg kg^−1^; and for IR36 of 230 and 290 mg kg^−1^. The high foliar [Zn] at e[CO_2_] might be suspected of bordering on toxic [35], yet they are associated with increased biomass, and particularly so for IR36 (Figure 1 and Figure 2). Therefore, it is tempting to suggest that a greater foliar [Zn] may be needed to satisfy the extra growth potential of rice at e[CO_2_]. This inference should be tested using multiple levels of Zn-supply and a larger number of rice cvv. In addition, for rice in the field, Zn-toxicity is only likely to occur when large excesses are applied, e.g., in contaminated wastes [36]. Interestingly, the observation that rice genotypes with a shorter grain filling period require less Zn [37], does not apply to IR26 and IR36 under our conditions.

Lastly, the time from germination to panicle emergence at marginal Zn-supply and a[CO_2_] was ~110 d for IR26 and ~90 d for IR36. For both cvv., the luxury Zn-supply and e[CO_2_] treatments separately decreased the time to panicle emergence by ~10 d, with a combined decrease of ~20 d. Since the time between panicle emergence and maturity was independent of the treatments (*p* > 0.05), the decreases in time to panicle emergence translated into decreased time to maturity. Reported e[CO_2_] effects on days to panicle emergence at are not consistent [37]; however, should effects such as those in our study occur in the field, they would increase flexibility in sequential multiple-cropping systems, and in the absence of other factors, the benefits would be expected to increase as the [CO_2_] increased [38].

### 2.3. Grain Yield and Composition

The treatment effects on grain yield plant^−1^ (Figure 3) are typically that: (1) IR36 out-yielded IR26; (2) within a cv. and a [CO_2_] treatment, luxury Zn increased yield—consequently, the maximum yield occurred for IR36 with luxury Zn at e[CO_2_]; (3) when both cvv. received the same Zn and CO_2_ treatments, the grain [Zn] of IR36 exceeded that of IR26—which is consistent with the inter-cultivar difference in the foliar Zn concentrations at panicle emergence (Figure 2), and with the inter-cultivar difference Zn-efficiency previously inferred from physiological parameters, such as chlorophyll content [25].

The pattern of grain yield responses to the treatments is complex (Figure 3). However, the number of panicles per plant is the principal driver of the inter-treatment differences in grain yield (R^2^ = 0.81, Figure 4). This relation is even stronger than that for vegetative biomass and tiller number at panicle emergence (Equations (1) and (2)). Such relations are not unusual [1].

Given that at panicle emergence luxury Zn-supply raised the foliar [Zn]—depending upon the cv. and the [CO_2_] treatment—from about 20–30 mg kg^−1^ to about 130–295 mg kg^−1^ (Figure 2), it is unsurprising that it also increased the grain [Zn] of both cvv. (Figure 5A). And, given the inter-cultivar difference in Zn-efficiency, it is unsurprising that IR36 typically had a greater grain [Zn] than IR26, and especially so at luxury-Zn (Figure 5A). Increasing the Zn-supply is expected to increase grain [Zn] [5,19,39], and such increases are important for dietary requirements in human populations for which rice is the staple carbohydrate [28,40,41]. Continuing with IR26 as the example, at marginal Zn-supply, e[CO_2_] decreased the grain [Zn] from 32 to 28 mg kg^−1^. A similar effect occurred for IR36 (Figure 5A), and depressions of grain [Zn] at e[CO_2_] are widely reported [42].

The grain [N] was greater in IR36 than in IR26 (Figure 5B). Grain [N] is universally reported to be depressed at e[CO_2_] [11,12,13,14,42,43], and that occurred at our marginal Zn-supply (*p* < 0.05, Figure 5B). In stark contrast, this effect was absent at luxury Zn-supply (*p* > 0.05, Figure 5B). We can only speculate about why this phenomenon may not have been observed previously and about the mechanism(s) that may underpin it: (1) the luxury Zn-supply caused a much greater foliar [Zn] at panicle emergence (Figure 2) than is commonly targeted in studies of Zn-sufficiency; and (2) it may be no coincidence that the major ligand involved in phloem transport of Zn in rice is the N-rich compound, nicotianamine [44], nor perhaps that Zn deficiency affects N-metabolism in the meristem of rice [45]. Lastly, soil Zn treatments can be costly and the efficacy of smaller foliar Zn applications in preserving the N concentration of grain at e[CO_2_] should be explored.

## 3. Materials and Methods

### 3.1. Plant Materials and Growth Conditions

Seeds of two rice cvv. (*Oryza sativa* subsp. indica) differing in Zn efficiency, IR36 (efficient) and IR26 (inefficient) [25], were obtained from the Australian Grains Gene Bank. Seeds were surface sterilised in sodium hypochlorite solution, washed with sterile water and germinated on moist filter paper in a Petri dish at 30 °C for 48 h in the dark. Four germinated seeds were planted per pot, the pots being 220 mm tall, 170 mm in diameter, and containing 1.25 kg of acid-washed sand. There were six replicate pots per cv. for each treatment. Half- strength nutrient solution [46] was supplied for 14 d, after which one seedling was retained per pot and the solution concentration was increased to full strength. Two different concentrations of Zn sulfate (~0.1 or ~6.5 µM Zn) were added to the nutrient solution to provide either a marginal or a luxury Zn-supply [47]. Nutrient solutions were topped up with reverse osmosis water every second day to compensate for evaporative and transpirational losses, and the solutions were replaced at ten-day intervals.

There were two concentrations of CO_2_, i.e., the local ambient concentration (a[CO_2_], ~400 µL L^−1^) in one greenhouse, and elevated (e[CO_2_], ~700 µL CO_2_ L^−1^) in the adjoining, identical greenhouse. This e[CO_2_] was chosen because it may be reached within the 21st Century [48]. It was achieved by injecting food grade CO_2_ (BOC Gas, Australia) into one of the greenhouses and continuous monitoring using an infra-red gas analyzer calibrated as recommended by the manufacturer (Guardian SP 97301, Edinburgh Instruments, United Kingdom). The greenhouses were located on the Toowoomba campus of the University of Southern Queensland (latitude 27°33′38.02″ S; longitude 151°55′55.20″ E; 690 m asl). The day/night conditions were 28/25 °C and the light period was natural (12–13 h d^−1^) and the season was spring through summer.

The time to first panicle emergence was recorded, and one set of pots was harvested on that day (*n* = 3); the remaining set was harvested at physiological maturity (*n* = 3). The plants were washed three times with reverse osmosis water then separated into roots, stems, leaves and panicles. The tillers and panicles were counted. The roots, stems, leaves and panicles were dried at 65 °C to constant weight and the masses (excluding panicles) recorded. The expression *vegetative biomass* refers to the sum of the masses of leaves, stems and roots. For each panicle, grain was dehusked manually, damaged kernels were removed, and we recorded the number of whole grains and their mass. For each plant, the whole grains were pooled and the weight of 100 was recorded and expressed as 1000 grain weights.

### 3.2. Postharvest Treatments and Measurements

Whole brown grain was finely ground using a porcelain mortar and pestle, and other dried plant parts were ground using a cyclonic mill (Cyclotec, Foss, Denmark). Dry subsamples of all parts (~300 mg) were digested using 3 mL of concentrated nitric acid and 2 mL of perchloric acid at 210 °C. Digests were diluted to 25 mL and analysed using inductively coupled plasma-optical emission spectrometry (Agilent Technologies, USA) for the following elements: sodium (Na), potassium (K), calcium (Ca), magnesium (Mg), manganese (Mn), iron (Fe), copper (Cu), Zn, phosphorus (P) and sulfur (S). Nitrogen was measured on a second subsample (~250 mg) by using a combustion analyzer (Model CN 628, LECO, Germany). Both instruments were operated following the manufacturer’s recommendations. Element concentrations were expressed on a dry weight basis.

### 3.3. Statistical Analysis

Data were subjected to analysis of variance (ANOVA) and principal components analysis (PCA) using Statistica (Version 13.5.0.17; TIBCO Software Inc.). Heteroscedastic data sets were appropriately transformed before ANOVA, and were standardised before PCA.

## 4. Conclusions

As a genetic resource, IR36 has advantages over IR26, such as: a greater mineral uptake; a shorter time to maturation; greater grain production; and a greater ability to increase grain Zn concentration in response to increased Zn supply. Importantly, luxury Zn-supply maintained grain [N] in both cvv. at e[CO_2_]. This novel observation may indicate an increased demand for Zn at e[CO_2_], with major implications for future rice yield and quality. Given these possibilities, our findings merit detailed investigation, including the: development of Zn dose-response relations for a wider range of rice genotypes and [CO_2_]; and evaluation of the relative agronomic benefits of soil versus smaller foliar applications of Zn and N. Finally, a mechanistic understanding of the findings might benefit consumers dependent on the nutritional value of rice and other C_3_ staples grown under increasing atmospheric CO_2_ concentrations.

## Figures and Tables

**Figure 1 plants-12-00839-f001:**
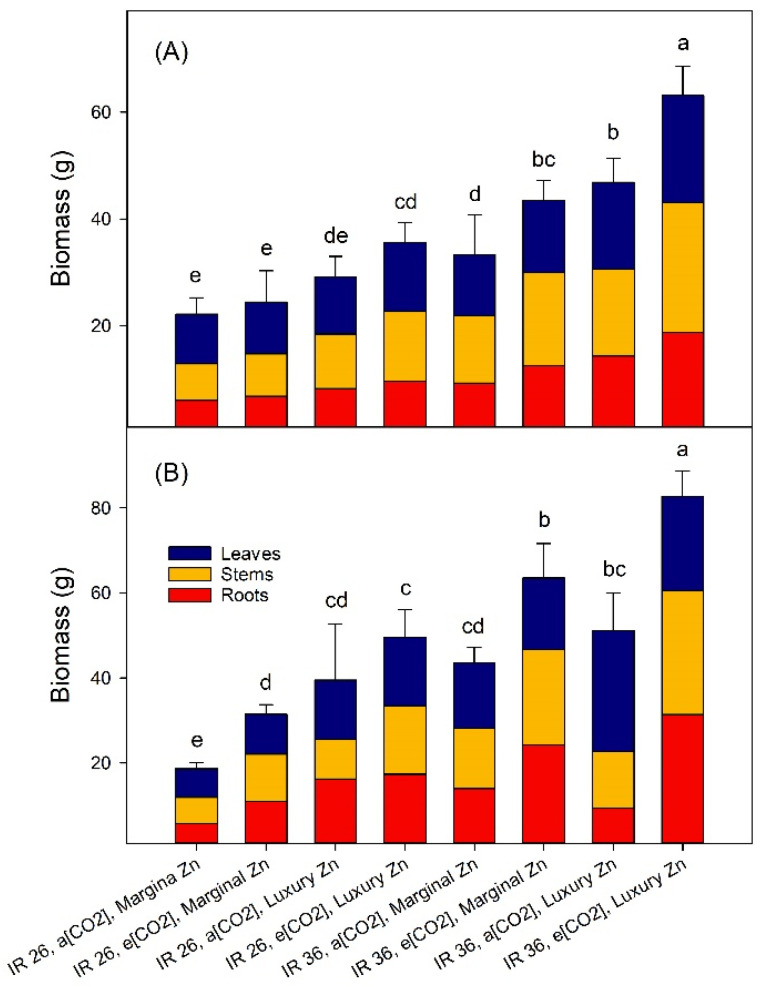
Treatment effects on the mass of leaves, stems and roots (vegetative biomass) at: (**A**) panicle emergence, and (**B**) maturity. Data are means of three replicates, the bars represent standard errors, and columns headed by different letters differ significantly (*p* < 0.05). For ANOVA see Appendix A.

**Figure 2 plants-12-00839-f002:**
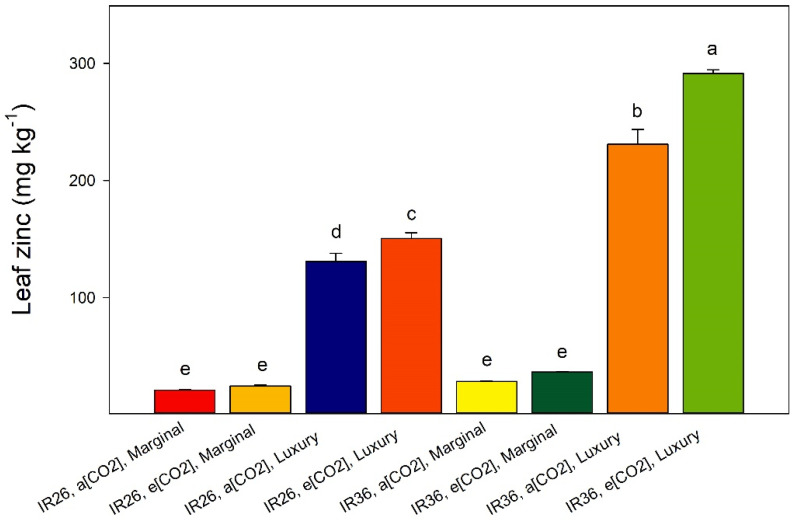
Foliar zinc concentration at panicle emergence. Data are means of three replicates, the bars represent standard errors, and columns headed by different letters differ significantly (*p* < 0.05).

**Figure 3 plants-12-00839-f003:**
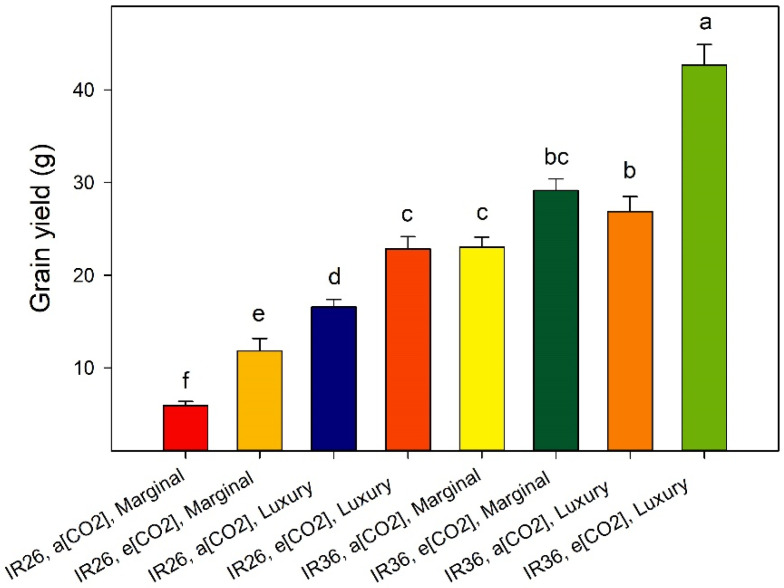
Treatment effects on grain yield. Values are means of three replicates, the bars represent standard errors, and columns headed by different letters differ significantly (*p* < 0.05).

**Figure 4 plants-12-00839-f004:**
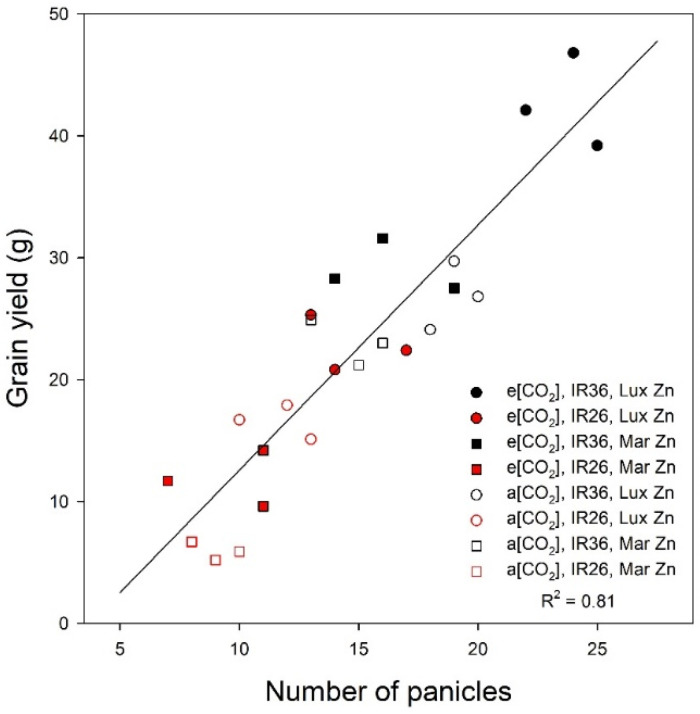
Relation between grain yield and panicle number at maturity. Data are for individual replicates (*n* = 3) of each of the eight (var × Zn × CO_2_) treatment combinations.

**Figure 5 plants-12-00839-f005:**
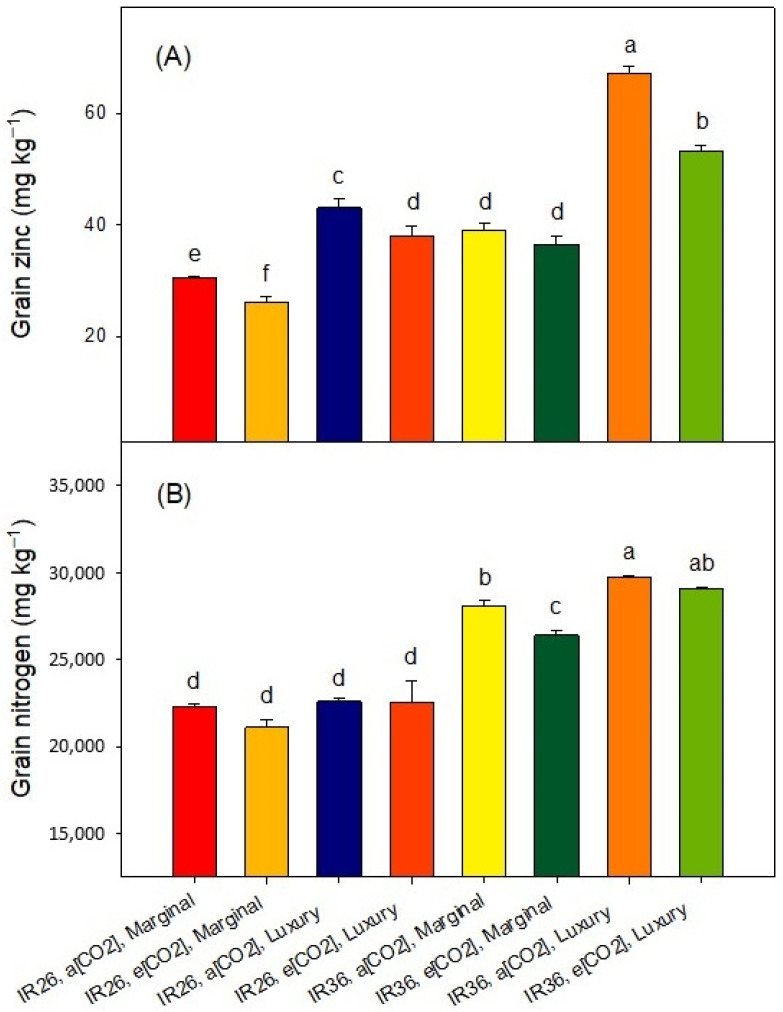
Treatment effects on the (**A**) [Zn] and (**B**) [N] in the grain. The values are means of three replicates, the bars represent standard errors, and columns headed by different letters differ significantly (*p* < 0.05). The ANOVA is presented in Appendix A.

## Data Availability

All data generated or analysed in this study are available within the manuscript or its Appendix A.

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
