# Peer review of "Luxury Zinc Supply Prevents the Depression of Grain Nitrogen Concentrations in Rice (*Oryza sativa* L.) Typically Induced by Elevated CO_2"

_plants, 2023, doi:10.3390/plants12040839_

Round 1

Reviewer 1 Report

The manuscript discuss the effect of luxury Zn supply on N content in rice cultivars under increased level of CO2. The MS is written very well. The results are well described and discussed in light of the existing knowledge. There are few suggestions that may be addressed before the MS is accepted for publication.

1. The symbols of N and Zn are unnecessarily written in brackets, which should be removed.

2. The concentration of N is given in mg kg-1, however, such higher values of N concentrations are usually presented in %.

3. The effect of luxury Zn supply on the concentration of other minerals may be discussed in the light of very recent article on rice cultivars by Naeem et al. (2022) ( https://doi.org/10.3390/agronomy12010049).

4. Fig. 1. Instead of staking up, I suggest presenting the biomass data of different plant parts in independent subfigures named as a, b, c, d, and so on. 

5. Figure 1, 2, 3 and 5: The data for the genotypes should be grouped into two groups, each for one genotype. The name of genotypes should be written in x-axis while treatment should be given in legend presented at the top within figure body. The statistical letters should also be written on bars in each figure. For guidance, please see Naeem et al. (2022) ( https://doi.org/10.3390/agronomy12010049).

Author Response

publication.

  1. The symbols of N and Zn are unnecessarily written in brackets, which should be removed.

The square brackets are used to indicate ‘concentration’ for CO2 and for the elements Zn and N to avoid frequent repetition of the word ‘concentration’ throughout the article. Consequently, we have retained the use of square brackets for this purpose.

  1. The concentration of N is given in mg kg-1; however, such higher values of N concentrations are usually presented in %.

We used mg kg-1 for all elements for simplicity and because this is an SI unit whereas % is not.  The alternative SI unit for larger concentrations is g kg-1, which we have now used for all the ‘major’ elements.

  1. The effect of luxury Zn supply on the concentration of other minerals may be discussed in the light of very recent article on rice cultivars by Naeem et al. (2022) (https://doi.org/10.3390/agronomy12010049). Agronomy202212(1), 49; https://doi.org/10.3390/agronomy12010049

We thank the Reviewer for drawing our attention to this recently accepted paper and have included it in our discussion.

  1. Fig. 1. Instead of stacking up, I suggest presenting the biomass data of different plant parts in independent subfigures named as a, b, c, d, and so on. 

We agree that the bars for the mass of all the vegetative components could be presented separately.  However, grain N not vegetative growth is our focus.  Therefore, we consider that the additional space required to present unstacked data would be a distraction.  Consequently, we have retained the stacks.

  1. Figure 1, 2, 3 and 5: The data for the genotypes should be grouped into two groups, each for one genotype. The name of genotypes should be written in x-axis while treatment should be given in legend presented at the top within figure body. The statistical letters should also be written on bars in each figure. For guidance, please see Naeem et al. (2022) (https://doi.org/10.3390/agronomy12010049).

The data are already grouped as the Reviewer suggests with the names of the genotypes shown on the X-axis (IR26 and IR36).  We added the requested statistical information to all the figures and added ANOVAS to the supplementary information.

Reviewer 2 Report

The paper by N. Nakandalage et al. entitled 'Luxury Zinc Supply Prevents the Depression of Grain Nitrogen Concentrations in Rice (Oryza sativa L.) Typically Induced by Elevated CO2' is an interesting study on the effect of zinc application on rice yield and grain nutritive mineral quality. The authors revealed that luxury zinc application increased foliar and grain zinc concentration in rice plants under pot experiments using hydroponics in comparing the situations of ambient and elevated atmospheric [CO2]. The obtained conclusion is that IR36, efficient in absorbing Zn, has an advantage in grain production and obtains more Zn in grains. Luxury zinc application increased not only Zn but also nitrogen in grains. These results are important for food nutrition for rice consumers, which are increasing now in the world. The method, analysis, and discussion are appropriate. However, I found some mistakes and some questions and mentioned them later.

1) The abstract of the paper should be written so that readers do not mistake that the experiment was done under a field and normal soil condition. In the abstract, it is informative to note that your experiment was done using pots and hydroponic with ash-washed sand.

2) In the keywords, it is important to write [CO2] or other words to explain that you changed [CO2].

3) On page 2, you note a nitrogen effect at the heading stage. But you did not change nitrogen. Nitrogen is not a factor in this experiment. It is better to change the word ‘nitrogen effect’ to another word.

4) You are confused with the two terms ‘panicle emergence’ and ‘panicle initiation. Panicle emergence, the appearance of the panicle, is the heading stage, which indicates the time when the panicle emerges and starts flowering. On the other hand, panicle initiation means young panicle primordium initiation, and it indicates the time when the young panicle differentiates and refers to about 25 days before heading. Please don't confuse the two words. And crop scientists prefer to use the word ‘heading’ rather than ‘panicle emergence’.

5) Although you note that analysis of variance (ANOVA) was performed in materials and methods, you did not note ANOVA results in the results. Please make tables for ANOVA. For the data in Figure 1, ANOVA results should indicate significant differences between treatments.

6) Please clearly state that vegetative weights did not include panicle dry weight in the materials and methods and notes in the tables and figures Data on panicle weight should also be provided.

7) You should design better colors for graphs. Especially the color coding in Figure 2 is strange. Readers cannot compare different treatments because different treatments share the same color.

8) Zinc is not very common as a fertilizer, and it is far from cheap. In this experiment, we believe that the effect of zinc in the pot is clearly apparent, but it is better to consider how much cost this luxuriously absorbed zinc level is needed and whether this level of Zn application is valid in the field of low-income countries because rice consumers in high-income countries do not need Zn from rice grains because they can take Zn from other foods.

9) It should be noted that rice plants were not damaged by too high concentrations of zinc in the soil. You should mention that this soil Zn concentration does not cause zinc damage. Zinc excess disorder occurs when soil Zn concentration is over 1000ppm.

10) Please show the significance of regression equations for Equations 1 and 2 on the fourth page. Please clearly write the number of replicates in the equations (4 or 12, 12 is valid when you treat all pots independently.

11) At the bottom of the fourth page, Figure 5A was not cited, and you mentioned physiological parameters. But what ‘physiological’ is this?

12) In page 5, you quoted Figure 4 but quoting Figure 4 is not appropriate here.

13) In the Reference list, in the number 28, Tadahiko and Koji are not family names for Japanese. They are first names. Please check the original paper. M. is Mae (family name), but I do not know who O. is. Mae is one of the famous plant nutrition scientists in Japan.

14) You did not clearly mention the method of yield determination. How did you calculate it using thousand grain weight? How did you measure them? You need panicle number and grain number per panicle. Please note them. It is better to not ripening percentage or seed set %.

15) There is no point in including data on minerals other than zinc in this paper. At the end of the results, there is foliar application vs. soil, but since this experiment was done under hydroponics, neither soil nor foliar application, you should cite papers if you want to mention this. Soil is a more complex system, and the results of hydroponics cannot be simply applied to rice production under field conditions.

16) Figures with principal component analysis are not necessary in this paper.

17) Zinc deficiency injury in rice is determined by soil pH, soil type, and other soil factors (Fundamentals of rice crop sciences, Yoshida, S. International Rice Research Institute, 1981). Please mention this in the study.

Author Response

1) The abstract of the paper should be written so that readers do not mistake that the experiment was done under a field and normal soil condition. In the abstract, it is informative to note that your experiment was done using pots and hydroponic with acid-washed sand.

We thank the Reviewer for this comment and have added the requested information

2) In the keywords, it is important to write [CO2] or other words to explain that you changed [CO2].

We thank the Reviewer for this comment and have amended the keywords

3) On page 2, you note a nitrogen effect at the heading stage. But you did not change nitrogen. Nitrogen is not a factor in this experiment. It is better to change the word ‘nitrogen effect’ to another word.

We thank the Reviewer for this comment and have included the words ‘The concentration of N in the medium was fixed.’ at the foot of the Introduction

4) You are confused with the two terms ‘panicle emergence’ and ‘panicle initiation. Panicle emergence, the appearance of the panicle, is the heading stage, which indicates the time when the panicle emerges and starts flowering. On the other hand, panicle initiation means young panicle primordium initiation, and it indicates the time when the young panicle differentiates and refers to about 25 days before heading. Please don't confuse the two words. And crop scientists prefer to use the word ‘heading’ rather than ‘panicle emergence’.

We thank the Reviewer for his specialized knowledge of rice development and have replaced the word ‘initiation’ by ‘emergence’ throughout.

5) Although you note that analysis of variance (ANOVA) was performed in materials and methods, you did not note ANOVA results in the results. Please make tables for ANOVA. For the data in Figure 1, ANOVA results should indicate significant differences between treatments.

As suggested by the Reviewer, the significance of the treatment effects is now shown in Figs 1 - 3 and 5. In addition key ANOVA tables have been added to the supplementary information.

6) Please clearly state that vegetative weights did not include panicle dry weight in the materials and methods and notes in the tables and figures Data on panicle weight should also be provided.

We have clearly stated that the mass of panicles (i.e., minus grain) was not recorded.  The panicle masses were very small relative to the other plant components.

7) You should design better colors for graphs. Especially the color coding in Figure 2 is strange. Readers cannot compare different treatments because different treatments share the same color.

We thank the Reviewer for this comment and have changed the colours and increased their intensity.

8) Zinc is not very common as a fertilizer, and it is far from cheap. In this experiment, we believe that the effect of zinc in the pot is clearly apparent, but it is better to consider how much cost this luxuriously absorbed zinc level is needed and whether this level of Zn application is valid in the field of low-income countries because rice consumers in high-income countries do not need Zn from rice grains because they can take Zn from other foods.

We agree that soil application of Zn in the field may be expensive; consequently, at the foot of the discussion have added the words    ‘Lastly, soil Zn treatments can be costly and the efficacy of smaller foliar Zn applications in preserving the N concentration of grain at e[CO2] should be explored‘.  The same idea is now also included in the conclusion and abstract

9) It should be noted that rice plants were not damaged by too high concentrations of zinc in the soil. You should mention that this soil Zn concentration does not cause zinc damage. Zinc excess disorder occurs when soil Zn concentration is over 1000ppm.

Good point thank you.  We have added the reference on toxicity by Kaur H, Garg N. 2021. Zinc toxicity un plants: a review. Planta 253: 129. https:/doi.org/10.1007/s00425-021-03642-z

10) Please show the significance of regression equations for Equations 1 and 2 on the fourth page. Please clearly write the number of replicates in the equations (4 or 12, 12 is valid when you treat all pots independently.

We thank the reviewer of this suggestion and have added the expression, n = 12

11) At the bottom of the fourth page, Figure 5A was not cited, and you mentioned physiological parameters. But what ‘physiological’ is this?

Fig 5A has been cited as suggested and chlorophyll content has been mentioned explicitly

12) In page 5, you quoted Figure 4 but quoting Figure 4 is not appropriate here.

We thank the Reviewer. This was an error and has been corrected.

13) In the Reference list, in the number 28, Tadahiko and Koji are not family names for Japanese. They are first names. Please check the original paper. M. is Mae (family name), but I do not know who O. is. Mae is one of the famous plant nutrition scientists in Japan.

We thank the Reviewer for picking up his error.  The Japanese names have now been abbreviated correctly.

14) You did not clearly mention the method of yield determination. How did you calculate it using thousand grain weight? How did you measure them? You need panicle number and grain number per panicle. Please note them. It is better to not ripening percentage or seed set %.

We thank the reviewer and have clarified in the methods exactly what was measured and calculated.

15) There is no point in including data on minerals other than zinc in this paper. At the end of the results, there is foliar application vs. soil, but since this experiment was done under hydroponics, neither soil nor foliar application, you should cite papers if you want to mention this. Soil is a more complex system, and the results of hydroponics cannot be simply applied to rice production under field conditions.

We respectfully disagree with the Reviewer.  The analysis for minerals other than Zn are essential to deflect criticism that our Zn response may have been due to an imbalance in some other essential mineral.  Therefore we have retained this supplementary data table. 

We agree that what happens in hydroponics may not occur in the field and have added words to that effect to both the Abstract and the Conclusions

16) Figures with principal component analysis are not necessary in this paper.

We agree and have deleted the PCA

17) Zinc deficiency injury in rice is determined by soil pH, soil type, and other soil factors (Fundamentals of rice crop sciences, Yoshida, S. International Rice Research Institute, 1981). Please mention this in the study.

We agree.  We have cited the suggested reference by Yoshida. 

Reviewer 3 Report

This manuscript investigate the possibility of applying Zinc to improve low Z and N concentration in polished rice under ambient and elevated CO2. The fond that luxury zinc supply could prevent the depression of grain N concentration in rice. The topic is interested. However, the paper is poorly organized in terms of structure arrangement and data statistic and presenting. In addition, the  citation of the figure in the text is very confusing. It seems that the author did not present their data in the manuscript.  I strongly suggest the author separate the part of RESULTS and DISCUSSION  section, as it is difficult to understand what is your finding and what is in the literature in the current form. For the experimental design, why not include a CONTROL without Zinc amendment?

Author Response

This manuscript investigate the possibility of applying Zinc to improve low Zn and N concentration in polished rice under ambient and elevated CO2. The fond that luxury zinc supply could prevent the depression of grain N concentration in rice. The topic is interested. However, the paper is poorly organized in terms of structure arrangement and data statistic and presenting. In addition, the  citation of the figure in the text is very confusing. It seems that the author did not present their data in the manuscript.  I strongly suggest the author separate the part of RESULTS and DISCUSSION  section, as it is difficult to understand what is your finding and what is in the literature in the current form. For the experimental design, why not include a CONTROL without Zinc amendment?

The Reviewer suggests that we separate the Results and Discussion to avoid confusion by readers.  Neither of the other two Reviewers had a problem understanding which results were ours and which were cited from the literature. 

As to a control with a zero-zinc addition, this was not done because the plants would have died.  That is, a zero-zinc treatment would have added no useful information.